https://doi.org/10.1038/s42003-019-0563-7　　**OPEN**

# Adipocyte metabolism is improved by TNF receptor-targeting small RNAs identified from dried nuts

Katia Aquilano[1], Veronica Ceci[1], Angelo Gismondi[1], Susanna De Stefano[2], Federico Iacovelli [1],
Raffaella Faraonio[3], Gabriele Di Marco[1], Noemi Poerio[1], Antonella Minutolo [1], Giuseppina Minopoli[3],
Antonia Marcone[3], Maurizio Fraziano[1], Flavia Tortolici[1], Simona Sennato[4], Stefano Casciardi[5], Marina Potestà[1],
Roberta Bernardini[6], Maurizio Mattei[1,6], Mattia Falconi[1], Carla Montesano [1], Stefano Rufini[1],
Antonella Canini[1] & Daniele Lettieri-Barbato[1,7]

There is a growing interest in therapeutically targeting the inflammatory response that underlies age-related chronic diseases including obesity and type 2 diabetes. Through integrative small RNA sequencing, we show the presence of conserved plant miR159a and miR156c in dried nuts having high complementarity with the mammalian TNF receptor superfamily member 1a (Tnfrsf1a) transcript. We detected both miR159a and miR156c in exosome-like nut nanovesicles (NVs) and demonstrated that such NVs reduce Tnfrsf1a protein and dampen TNF-α signaling pathway in adipocytes. Synthetic single-stranded microRNAs (ss-miRs) modified with 2′-O-methyl group function as miR mimics. In plants, this modification naturally occurs on nearly all small RNAs. 2′-O-methylated ss-miR mimics for miR156c and miR159a decreased Tnfrsf1a protein and inflammatory markers in hypertrophic as well as TNF-α-treated adipocytes and macrophages. miR156c and miR159a mimics effectively suppress inflammation in mice, highlighting a potential role of plant miR-based, single-stranded oligonucleotides in treating inflammatory-associated metabolic diseases.

[1] Department of Biology, University of Rome Tor Vergata, Rome, Italy. [2] Department of Chemical Sciences and Technologies, University of Rome Tor Vergata, Rome, Italy. [3] Department of Molecular Medicine and Medical Biotechnologies, University of Naples Federico II, Naples, Italy. [4] CNR-ISC and Department of Physics, Sapienza University of Rome, Piazzale A. Moro 2, 00185 Rome, Italy. [5] Department of Occupational and Environmental Medicine, Epidemiology and Hygiene, National Institute for Insurance against Accidents at Work (INAIL) Research, Rome, Italy. [6] Interdepartmental Service Center-Station for Anima Technology (STA), University of Rome Tor Vergata, Rome, Italy. [7] IRCCS Fondazione Santa Lucia, 00143 Rome, Italy. Correspondence and requests for materials should be addressed to K.A. (email: katia.aquilano@uniroma2.it) or to D.L-B. (email: d.lettieribarbato@hotmail.it)

A persistent increase of circulating levels of tumor necrosis factor-alpha (TNF-α) occurring during obesity or aging has an important role in pathogenesis of systemic insulin resistance[1–3]. Several works identified that TNF-α receptor type 1 (Tnfrsf1a)-induced signals are required and sufficient to impair insulin action[4–6]. Several downstream mechanisms have been proposed by which TNF-α might cause insulin resistance both in white and brown adipocytes[7]. In particular, TNF-α affects the expression of insulin receptor, insulin receptor substrate-1, and glucose trasporter type 4, which are involved in the insulin-stimulated glucose uptake in adipocytes[6,8,9]. Furthermore, TNF-α also potently stimulates lipolysis and this may contribute to catabolic disease states such as the cachectic conditions associated with cancer[10]. Accordingly, inhibition of lipolysis pathway ameliorates the severity of cachexia improving cancer prognosis[11]. Aberrant TNF-α signaling cascade is also observed in several autoimmune diseases (e.g. rheumatoid arthritis)[12]. Therefore, finding efficient strategies countering inflammation is a promising field in drug discovery.

The mechanisms of action of microRNAs (miRs) have led to their suggestions for therapeutic applications in analogy with exogenously supplemented small interfering RNAs (siRNAs), which induce degradation of sequence-specific homologous mRNAs via RNA interference (RNAi). Single-stranded miRs (ss-miR) containing the seed sequence mimics the ability of duplex miRs in inhibiting the expression of target genes[13–15] and the chemical modification with 2′-O-methyl at 3′-end increases their stability in vivo and does not affect their biological activity[16]. Notably, 2′-O-methyl modification is a peculiarity of plant miRs[17]. Several reports demonstrated that plant miR mimics target mammalian genes, improving inflammation[18] and cancer[19].

In this work, by high throughput sequencing and computational predictions we identified conserved plant miRs in dried nuts that have an anti-inflammatory potential. Functional studies revealed that synthetic single-stranded miR mimics for conserved plant miR156c and 159a are able to inhibit the TNF-α signaling pathway through the targeting of Tnfrsf1a gene transcript in experimental models of obesity. The data here presented point to the use of plant miR-based single-stranded oligonucleotides for the treatment of chronic low-grade inflammatory states such as those observed during obesity.

## Results

### Small RNAs isolated from dried nuts limit inflammatory response and enhance insulin-mediated glucose uptake in hypertrophic adipocytes. Excessive calorie intake leads to formation of hypertrophic adipocytes promoting senescence-like changes, such as increased production of pro-inflammatory cytokines[20–22]. Accordingly, analysis of a publicly available microarray dataset (GEO: GSE32095) revealed that in white adipose tissue of mice treated with high-fat diet (HFD) about 6200 genes and 8700 genes are up-regulated and down-regulated, respectively (Fig. 1a). Among the up-regulated genes an enrichment of inflammatory- and hypoxia responses (4.1%, $p < 0.001$ and 3.5%, $p < 0.001$, respectively) were found (Fig. 1b).

Small non-coding RNAs (sRNAs) are one of the most promising tools to treat a number of human diseases[23]. In recent years, the capacity of exogenous plant sRNAs to modulate mammalian gene transcripts and inflammatory pathways has emerged[19,24]. Based on the widely reported anti-inflammatory role of dried nuts[25], here we have hypothesized that sRNAs extracted from nuts could affect inflammatory signaling pathway in in vitro models of obesity-related adipose tissue inflammation. We used differentiated 3T3-L1 adipocytes at day 6 treated with TNF-α to test the effects of sRNAs derived from several nut specimens (i.e. *J. californica, C. avellana, J. regia*) on the inflammatory response. Pathological expansion of visceral adipose tissue triggers a hypoxia state that in turn elicits an inflammatory response[26]. Hence, differentiated 3T3-L1 adipocytes treated with a hypoxia-mimicking drug, such as cobalt chloride ($CoCl_2$), were also used. Interestingly, nut sRNAs restrained phospho-activation of p65-NFkB (p-NFkBp65) protein (Fig. 1c) and TNF-α gene expression (Fig. 1d) in both TNF-α- and $CoCl_2$-treated adipocytes compared to cells transfected with a negative small RNA [(−)sRNA]. Similar results were observed by transfecting TNF-α-treated T37i brown adipocytes with nut sRNA (Supplementary Fig. 1).

We also investigated the effects of nut sRNAs on differentiated 3T3-L1 adipocytes at day 16, another condition reproducing a hypertrophic and inflamed phenotype[22]. Compared to adipocytes at day 6, adipocytes at day 16 showed an accumulation of intracellular triglycerides (Fig. 1e) accompanied by the development of a secretory phenotype, as demonstrated by increased mRNAs levels of inflammatory cytokines (i.e. TNF-α, IL-1β, and IL-6) and production and release of TNF-α protein (Fig. 1f, g). We found that hypertrophic adipocytes transfected with the sRNAs extracted from nuts exhibited reduced mRNA levels of inflammatory cytokines (Fig. 1f) as well as decreased intracellular protein content and extracellular release of TNF-α (Fig. 1g) compared to the negative control.

The increased production of inflammatory cytokines has been implicated in the development of insulin resistance[27]. Accordingly, both hypertrophic (day 16) and TNF-α-treated 3T3-L1 adipocytes showed decreased levels of p-Akt (Fig. 1h), a marker of insulin sensitivity, and reduced glucose uptake (Fig. 1i) upon insulin stimulation. Transfection with nut sRNAs prevented p-Akt decrease (Fig. 1h) and enhanced glucose uptake in hypertrophic adipocytes (Fig. 1i). We also transfected adipocytes with sRNAs derived from fresh apple (*M. domestica*) and found that they have lower effectiveness in reducing inflammatory cytokines mRNA expression (only a decrease of IL-1β was evidenced) and increasing glucose uptake than sRNA isolated from dried nuts (Fig. 1f, i). Overall, these findings suggest that nut sRNAs may be useful to prevent inflammation and ameliorate insulin sensitivity in adipocytes.

### Small RNA-sequencing reveals conserved plant miRs in nuts that putatively target Tnfrsf1a. MicroRNAs (miRs) are the best-characterized sRNAs able to modulate gene expression in target cells. In order to evaluate miRs as possible mediators of the anti-inflammatory role observed with nut sRNAs, we moved at sequencing the sRNAs in our samples, and the miRNome profiling was then determined by their read frequencies in the libraries. In Fig. 2a–c, the read counts of the most abundant miRs detected in *J. regia* (Fig. 2a, red bars), *J. californica* (Fig. 2b, blue bars), and *C. avellana* (Fig. 2c, green bars) were reported. Interestingly, the Juglandaceae showed a richness in terms of miR families' abundance and their relative total quantities (Fig. 2a, b) similar to *C. avellana* (Fig. 2c). Through Venn diagram obtained by comparing our datasets, we observed an overlap of 185 miR families between the two Juglandaceae (Fig. 2d). Among these overlapping families, 29 were in common with *C. avellana* (Fig. 2d). Interestingly, 12 out of 29 common miRs (miRNuts) were also disclosed in other small RNA-seq deriving from unprocessed vegetable foods including apple, sweet orange, and grape wine (Fig. 2e, Table 1). Successively, we predicted the members of the conserved miR families in nut specimens (Table 1), and their interactions with gene transcripts mapping the Tnf signaling pathway were evaluated through computational

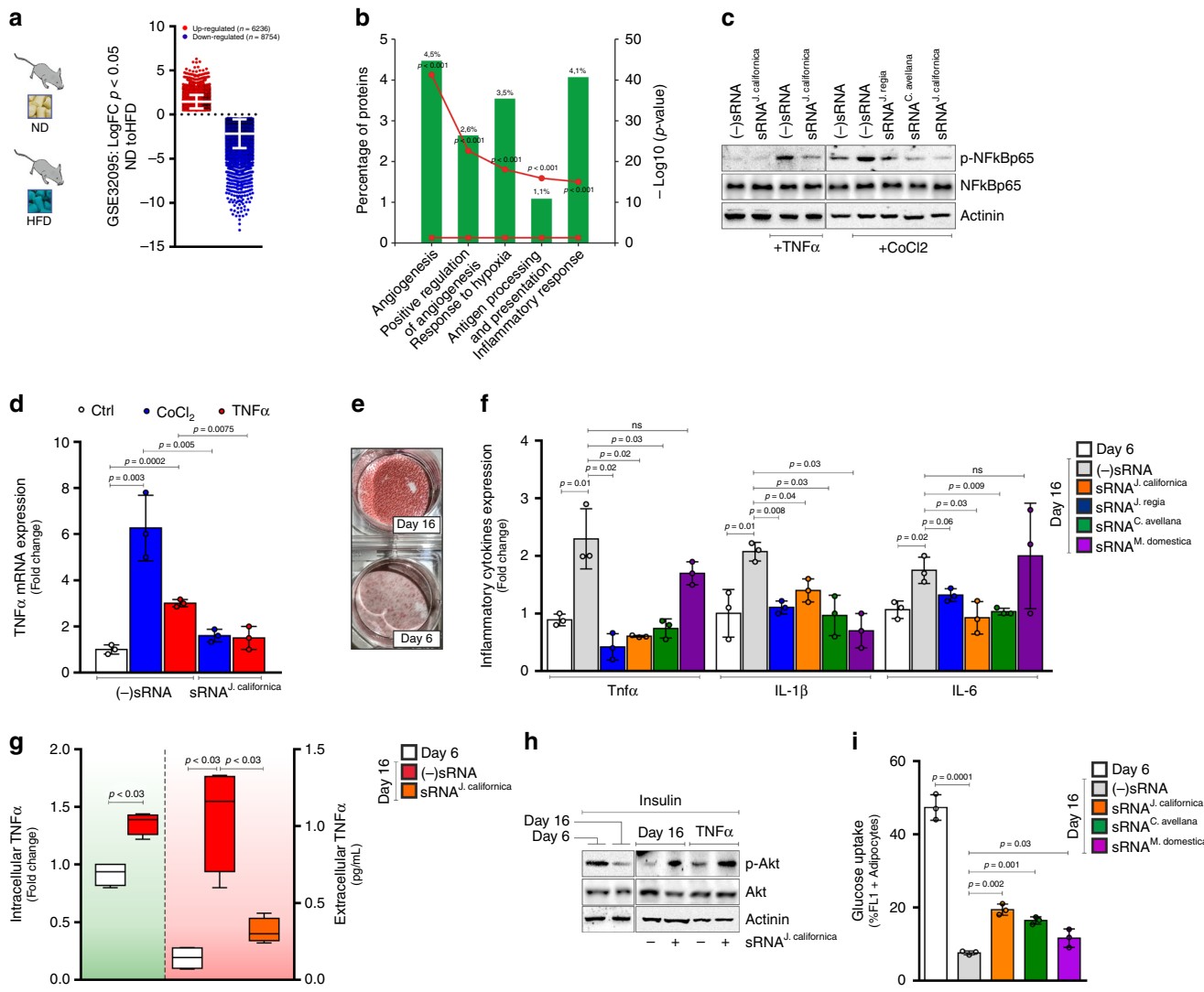

**Fig. 1** Small RNA isolated from nuts reduce cytokine expression and enhance insulin-mediated glucose uptake in adipocytes. **a**, **b** Transcriptomics data of differentially expressed genes ($p < 0.05$) obtained from Gene Expression Omnibus (GEO) dataset GSE32095 (**a**). Functional enrichment analysis of up-regulated genes (>2-fold change; $p < 0.05$) (**b**). **c**, **d** p-NFkBp65 protein (**c**) and TNF-α mRNA expressions (**d**) were analyzed in 3T3-L1 adipocytes treated with TNF-α or CoCl₂ and transfected with small RNA (sRNA) extracted from *J. californa*, *J. regia*, or *C. avellana*. Transfection with a scramble small RNA [(−)sRNA] was used as a negative control. Uncropped images are shown in Supplementary Fig. 6. **e** Accumulation of triglycerides was evaluated in normal (day 6) and hypertrophic (day 16) 3T3-L1 adipocytes by Oil red-O staining. **f**, **g** Cytokines mRNA expression (**f**), intracellular (**g**, left panel), and extracellular (**g**, right panel) TNF-α protein levels were analyzed in normal (day 6) and hypertrophic (day 16) adipocytes transfected with sRNA extracted from *J. californica*, *J. regia*, *C. avellana*, or *M. domestica* by qPCR, flow cytofluorimetry, and ELISA assays, respectively. **h** Akt-p(Ser473) was measured in insulin-stimulated hypertrophic (day 16) or TNF-α-treated adipocytes and transfected with sRNA extracted from *J. californa*. Uncropped images are shown in Supplementary Fig. 6. **i** Glucose uptake was measured by flow cytofluorimetry in insulin-stimulated hypertrophic adipocytes and transfected with sRNA extracted from *J. California*, *C. avellana*, and *M. domestica*. All immunoblots reported are representative of three independent experiments giving similar results. Actinin was used as a loading control. Data are expressed as means ± SD ($n = 3$)

analysis (IntaRNA v2.0). To strengthen the biological significance of Tnf signaling in adipose cells, the conservation rate of this pathway was dissected by CyKEGGParses[28]. As reported in Fig. 2f, among 35 nodes characterizing the Tnfrsf1a-mediated Tnf signaling pathway, 28 were maintained after tuning in adipocytes (green box). Of interest, the absence of seven nodes (yellow box) did not affect the Tnf signaling cascade in human adipose cells.

The computational analyses predicted that miR156c and miR159a can target members of Tnf inflammatory pathway in mammals, including humans (Tables 2 and 3). Then putative mRNA targets of the above miRs were filtered for their abundance in adipose tissues and adipocytes by using gene expression profiles (http://biogps.org). The bioinformatics

analysis revealed that Tnfrsf1a mRNA has an abundant expression in white (red arrow) and brown adipose tissue (blue arrow) as well as in 3T3-L1 adipocytes (green arrow) (Supplementary Fig. 2). Computational analyses on human TNFRSF1A gene transcript also predicted the presence of seed regions in plant miR156c and miR159a (Table 3). These results pointed to the mammalian Tnfrsf1a transcript as a putative target of miR159a and miR156c in adipocytes.

**miR-containing plant nanovesicles ameliorate insulin sensitivity and inflammation in in vitro and in vivo models of adipose tissue inflammation.** Interestingly, plant nanovesicles

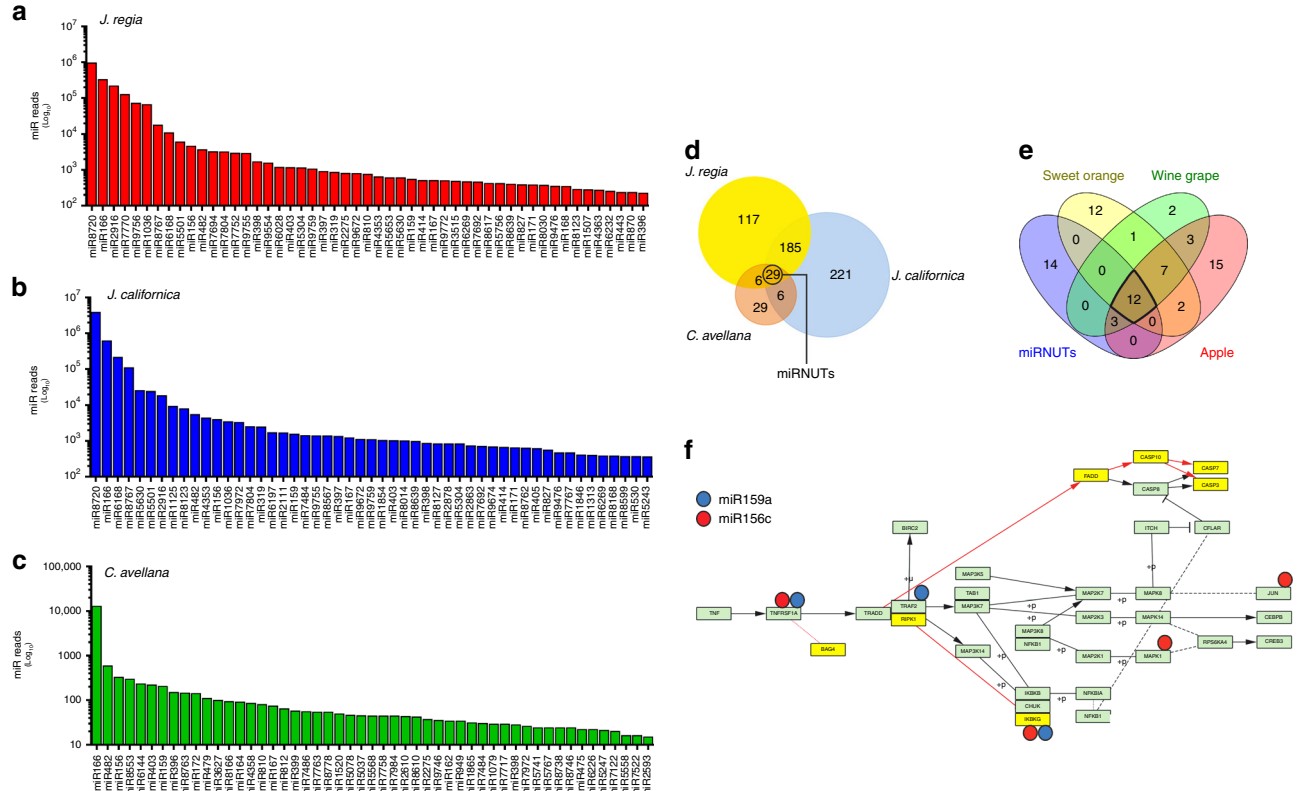

**Fig. 2** Small RNA-sequencing and computational target predictions. **a–c** Number of reads were obtained by small RNA sequencing in whole homogenates of *J. regia* (**a**), *J. californica* (**b**), and *C. avellana* (**c**). **d, e** The logical relationship between miR profiling obtained from our dataset (**d**) and other small RNA-seq (**e**) was evaluated by the Venn diagram. **f** Prediction of miR targets was carried out on KEGG Tnf signaling pathway using IntaRNA v2.0. Red and blue circles indicate high energy miR-mRNA interactions ($\leq -10$ kcal/mol). Green boxes represent the conserved nodes in adipocytes as evaluated through CyKEGGParses

(NVs) are proposed as tools to enhance miR delivery both in vitro and in vivo systems[29–31]. Indeed, plant NVs have been demonstrated to be resistant to gastric and intestinal phase of digestion process[32]. Herein we isolated nut NVs (Fig. 3a) and transmission electron microscopy analysis was initially carried out to verify their integrity and size. In Fig. 3b, we reported representative images showing a dimension that was around 200 nm. NVs were further analyzed by dynamic light scattering (Fig. 3b), which revealed an average diameter of 230 nm for *J. californica*: a diameter of about 90 and 330 nm for *J. regia* and a diameter of about 90 and 560 nm for *C. avellana*. To map the structural feature of nut NVs, gas chromatography mass spectrometry analysis was also performed and, as reported in Supplementary Table 1, lipids-like molecules were the predominant detected species. Finally, we observed that overall analyzed NVs showed a negative ξ-potential value ranging from −21 to −23 mV. According to previously reported data[24,32], our findings suggest that NVs purified from dried nuts display mammalian exosome-like features.

Next, we stained NVs with a selective RNA probe and through cytofluorimetric analysis we detected a similar RNA abundance in NVs of all the nut samples (Fig. 3c). NVs stained with the fluorescent RNA probe were tested for their ability to enter inside the cell. An increase of RNA-derived signal was detected by fluorescence microscopy (Supplementary Fig. 3a) and cytofluori-metric analysis (Supplementary Fig. 3b) in 3T3-L1 adipocytes. Notably, we found that miR159a and miR156c were present in overall the purified nut NVs along with other plant miRs such as miR319a, miR482b, miR396c, miR162a, miR166i, and miR167–5p (Supplementary Fig. 3c). Thus, we evaluated if NVs were effective in reducing the inflammatory profile in

hypertrophic adipocytes (day 16). As reported in Fig. 4a, overall NVs specimens were able to reduce TNF-α, Il-1β, and Il-6 mRNA expression as well as p-NFkBp65 protein (Fig. 4b) in adipose cells. As expected, nut NVs also lowered the intracellular TNF-α protein levels in TNF-α and CoCl$_2$-treated 3T3-L1 adipocytes (Fig. 4c, d). In line with the reduced production of cytokines, nut NVs enhanced glucose uptake in TNF-α treated adipocytes (Fig. 4e), suggesting their ability to improve insulin sensitivity. Notably, the same amount of NVs isolated from fresh apple were ineffective in reducing the mRNA level of pro-inflammatory cytokines in hypertrophic adipocytes (Fig. 4a), in line with the lowest amount of miR156c and miR159a detected in such NVs (Supplementary Fig. 3c).

Based on the in vitro findings, which demonstrated the capacity of nut NVs to down-regulate TNF-α signaling, we next moved at analyzing the effects of such NVs in an in vivo model of diet-induced obesity. In particular, we attempted at exploring the efficacy of nut NVs in reducing inflammation in mouse visceral white adipose tissue during HFD (60% kcal from fats for 16 weeks). NVs supplementation in combination with HFD (HFD + NV) resulted in ameliorated fasting glycemia and reduced serum alanine aminotransferase (Fig. 5a). In line with increased glucose uptake observed in vitro (Fig. 1i), we demonstrated an improved glucose tolerance in the HFD + NV group (Fig. 5b). In the HFD + NV group we also found an increase in total body and fat weight (Fig. 5c, d). In particular, while visceral adipose tissue remained unchanged, accumulation of brown and subcutaneous adipose depots was observed (Fig. 5d), likely due to adipocyte hypertrophy. Actually, an enlarged adipocyte size was observed in subcutaneous adipose tissue of

**Table 1 miR family members in nut specimens (http://www.mirbase.org)**

| miR family | J. californica | J. regia | C. avellana | Conserved miR family |
|---|:---:|:---:|:---:|:---:|
| >miR156c UUGACAGAAGAGAGAGAGCAC | ✓ | ✓ | ✓ | ° |
| >miR159a UUUGGAUUGAAGGGAGCUCUA | ✓ | ✓ | ✓ | ° |
| >miR162a UCGAUAAACCUCUGCAUCCAG | ✓ | | ✓ | |
| >miR166a UCGGACCAGGCUUCAUUCCCC | ✓ | | ✓ | ° |
| >miR167-3p AGAUCAUGUGGCAGUUUCACC | ✓ | ✓ | | ° |
| >miR167h UGAAGCUGCCAGCAUGAUCUUA | ✓ | ✓ | | ° |
| >miR168a-5p UCGCUUGGUGCAGGUCGGGAA | ✓ | ✓ | | |
| >miR171a UUGAGCCGCGUCAAUAUCUCC | ✓ | ✓ | ✓ | ° |
| >miR171b UUGAGCCGCGUCAAUAUCUCC | ✓ | ✓ | | ° |
| >miR172b GCAGCACCAUUAAGAUUCAC | ✓ | ✓ | | ° |
| >miR319a UUGGACUGAAGGGAGCUCCCU | ✓ | ✓ | ✓ | ° |
| >miR319f AUUGGACUGAAGGGAGCUCC | ✓ | | | ° |
| >miR395i CUGAAGUGUUUGGAGGAACUC | ✓ | | | |
| >miR396b-5p UUCCACAGCUUUCUUGAACUU | ✓ | | | ° |
| >miR398b UGUGUUCUCAGGUCGCCCCUG | ✓ | | | ° |
| >miR399a CGCCAAAGGAGAGUUGCCCUU | ✓ | | | ° |
| >miR403-3p UUAGAUUCACGCACAAACUCG | ✓ | ✓ | ✓ | ° |
| >miR408-3p AUGCACUGCCUCUUCCCUGGC | ✓ | | | |
| >miR482 UCUUUCCUACUCCUCCCAUUCC | ✓ | ✓ | | ° |
| >miR482c-5p GGAAUGGGCUGUUUUGGGAUG | ✓ | ✓ | | ° |
| >miR530 UGCAUUUGCACCUGCACCUCU | ✓ | | | |
| >miR827-5p UUUGUUGAUGGUCAUCUAUUC | ✓ | | | |
| >miR2916 UUGGGGGCUCGAAGACGAUCAGAU | ✓ | | | |
| >miR5653 GAGUUGAGUUGAGUUGAGUUGAGA | ✓ | ✓ | | |
| >miR9560-5p AGGCGGUGGAACAAAUAUGAACUU | ✓ | | | |
| >miR156d UGACAGAAGAGAGUGAGCAC | | | ✓ | ° |
| >miR162a-3p UUAGAUUCACGCACAAACUCG | | ✓ | | |
| >miR166d UCGGACCAGGCUUCAUUCCCC | | ✓ | | |
| >miR166f UCUCGGACCAGGCUUCAUUCC | | | ✓ | ° |
| >miR167f-5p UGAAGCUGCCAGCAUGAUCUU | | | ✓ | ° |
| >miR390a-5p AAGCUCAGGAGGGAUAGCGCC | | ✓ | | |
| >miR394a UUGGCAUUCUGUCCACCUCC | | ✓ | | |
| >miR395d CUGAAGUGUUUGGGGGGAACUC | | ✓ | | |
| >miR396a-5p UUCCACAGCUUUCUUGAACUG | | ✓ | | ° |
| >miR482b UCUUUCCUAUCCCUCCCAUUCC | | | ✓ | ° |
| >miR827 UUUGUUGAUGGUCAUCUAAUC | | ✓ | | |
| >miR829-3p UGCAUCAGUUGGUAUCAGAGCUCA | | | ✓ | |
| >miR1863b UUUGGCUCUGAUAACCAUGUUAAAU | | | ✓ | |
| >miR7758-5p UUAACGGUCAACUAACGGAUGGAC | | | ✓ | |
| >miR7782 CCUCUGCUCUGAUACCAUGU | | ✓ | | |

**Table 2 Computational analysis to predict the interactions between plant miRs and murine gene transcripts mapping for Tnf signaling pathway**

| Murine gene transcript (Transcript ID) | Mature miR sequence | Interaction energy (kcal/mol) |
|---|---|---|
| Tnfrsf1a (NM_011609.4) | >miR156c UUGACAGAAGAGAGAGAGCAC> miR159a UUUGGAUUGAAGGGAGCUCUA | −13.303 −12.0495 |
| Traf2 (NM_009422.3) | >miR159a UUUGGAUUGAAGGGAGCUCUA | −10.0283 |
| IkBkG (NM_001136067.2) | >miR159a UUUGGAUUGAAGGGAGCUCUA> miR156c UUGACAGAAGAGAGAGAGCAC | −11.5712 −12.276 |
| Mapk10 (NM_001318131.1) | >miR156c UUGACAGAAGAGAGAGAGCAC | −16.757 |
| Mapk1 (NM_001357115.1) | >miR156c UUGACAGAAGAGAGAGAGCAC | −13.002 |

**Table 3 Computational analysis to predict the interactions of miR156c and 159a with human TNFRSF1A gene transcript**

| Human gene transcript (Transcript ID) | Mature miR sequence | Interaction energy (kcal/mol) |
|---|---|---|
| TNFRSF1A (NM_001065.3) | >miR156c UUGACAGAAGAGAGAGAGCAC> miR159a UUUGGAUUGAAGGGAGCUCUA | −11.3401 −7.554 |

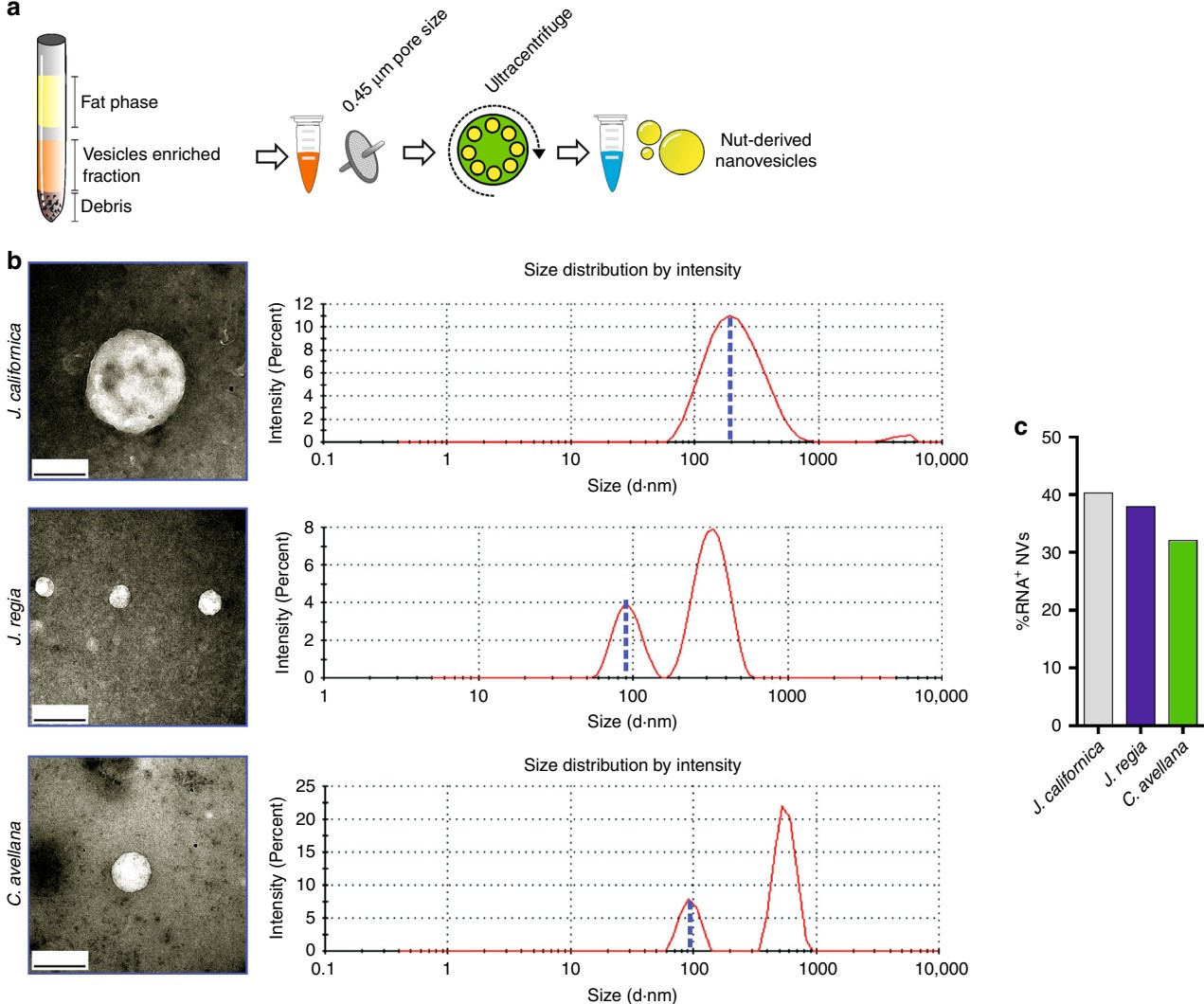

**Fig. 3 Isolation and characterization of plant NVs. a** Schematic representation of plant NVs isolation of *Juglans regia, Juglans californica, Corylus avellana, Bertholletia excelsa*, and *Malus domestica*. **b** Representative images obtained through transmission electron microscopy (left panels) and intensity-weighted size distribution calculated by dynamic light scattering (right panels) of NVs isolated from *J. californica, J. regia*, and *C. avellana*. Scale bars: 200 nm. **c** Isolated NVs were stained with an RNA-specific probe and the percentage of positive NVs was calculated by cytofluorimetry. The histogram reported is from one experiment representative of three giving similar results

the HFD + NV group (Supplementary Fig. 4a). Accordingly, an increased level of intracellular triglycerides was detected in T37i adipocytes treated with nut NVs (Supplementary Fig. 4b). The HFD + NV group also showed reduced TNF-α mRNA levels in adipose depots (Fig. 5e) and through cytokine antibody array, we confirmed an improved inflammatory profile in visceral adipose tissue of mice treated with HFD + NV (Fig. 5f, g).

**miR159a and miR156c target Tnfrsf1a gene transcript**. By inquiring transcriptome data (GSE32095), we profiled gene expression of TNF-α receptors (Tnfr) in visceral adipose tissue of HFD-treated mice. As expected, HFD significantly ($p < 0.05$) up-regulated Tnfr-related gene transcripts including Tnfrsf1a (Supplementary Fig. 4c). Differential gene expression between white and brown adipose tissue (GSE8044) showed that Tnfrsf1a is more abundantly expressed in white compared to brown adipose tissue (LogFC 1.36; $p < 0.001$).

We asked whether plant NVs were effective in modulating Tnfrsf1a level in adipocytes and in obese mice. Remarkably, NVs reduced Tnfrsf1a protein levels both in hypertrophic adipocytes

(Fig. 6a) as well in visceral adipose tissue of HFD-treated mice (Fig. 6b). Given that, among the miRs present in the plant NVs, miR156c and miR159a were predicted to target Tnfrsf1a mRNA, we asked whether these miRs were able to down-modulate Tnfrsf1a expression similarly to NVs.

ss-miR mimics might combine the power of function through the RNAi pathway with the more favorable pharmacological properties of single-stranded oligonucleotides[13–15]. Plant miRs have a naturally occurring 2′-O-methyl group that confers high stability to RNA oligonucleotides without affecting their biological activities[33]. Keeping in mind this aspect, we designed ss-miR mimics for plant miR156c and miR159a, containing 2′-O-methylation at 3′-end, and then transfected them in 3T3-L1 (day 16) and T37i adipocytes. The transfection of ss-miR156c mimic was also performed in human type 1 macrophages (M1) and TNF-α-stimulated murine RAW 264.7 cells, since during obesity macrophages recruitment is elicited contributing to inflammation and insulin resistance[34]. As transfection controls, a conventional murine negative small RNA interference [(−) sRNA] or single-stranded 2′-O-methylated miR167h mimic, which does not show any putative interactions with murine and

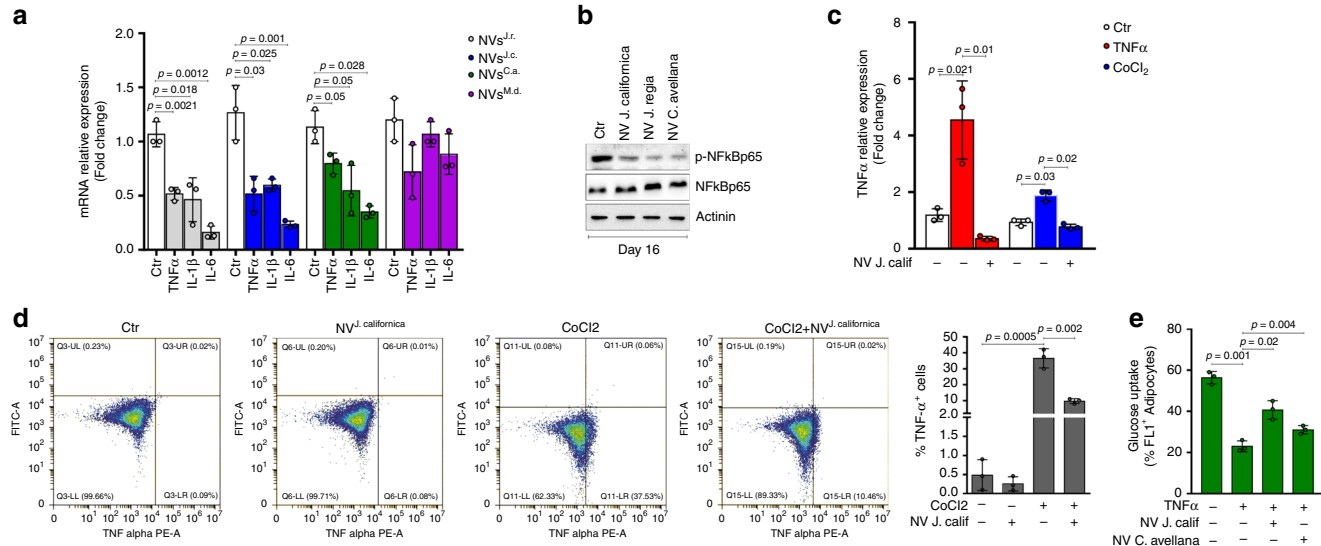

**Fig. 4** NVs reduce TNF-α levels and increase glucose uptake in adipocytes. **a** Cytokines mRNA expression was analyzed through RT-qPCR in hypertrophic adipocytes treated with NVs isolated from *Juglans regia* (J.r.), *Juglans californica* (J.c.), *Corylus avellana* (C.a.), and *Malus domestica* (M.d.). **b** p-NFkBp65 protein levels were analyzed in hypertrophic adipocytes treated with NVs isolated from *Juglans regia*, *Juglans californica*, and *Corylus avellana*. Immunoblots reported are representative of three independent experiments. Actinin was used as a loading control. Uncropped images are shown in Supplementary Fig. 7. **c**, **d** TNF-α mRNA expression (**c**) and intracellular TNF-α protein (**d**) levels were analyzed by RT-qPCR and cytofluorimetry, respectively, in adipocytes treated with TNF-α or CoCl₂ in combination with NVs isolated from *Juglans californica*. The gating strategy of cytofluorimetric analysis is shown in Supplementary Fig. 8. **e** Glucose uptake was measured by flow cytofluorimetry in insulin-stimulated hypertrophic adipocytes treated with NVs isolated from *Juglans californica* and *Corylus avellana*. Data are expressed as means ± SD (*n* = 3)

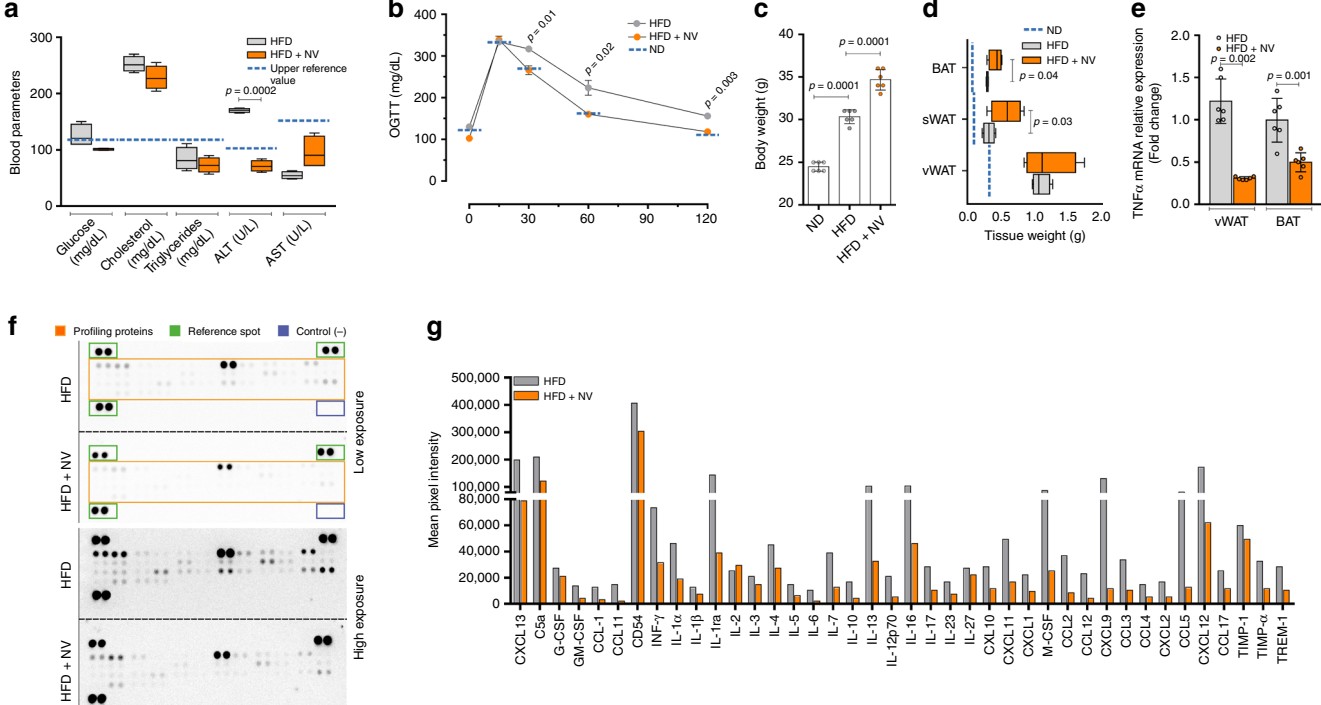

**Fig. 5** NVs improve inflammatory and metabolic profile in obese mice. **a** Biochemical parameters were analyzed in blood samples of mice fed with high-fat diet (HFD) or HFD supplemented with NVs isolated from *J. californica* (HFD + NV). Reference values were reported as blue dashed lines. **b** Oral glucose tolerance test (OGTT) was performed in mice fed with HFD or HFD supplemented with NVs (HFD + NV). Blood samples were collected at several time points from glucose administration. Values of mice fed with normal diet (ND) were reported as blue dashed lines. **c**, **d** Total body (**c**) and fat mass (**d**) weights of mice fed with HFD or HFD + NV. ND: mice fed with normal diet. **e** TNF-α mRNA expression in vWAT and BAT of mice fed with HFD or HFD + NV. **f**, **g** Cytokines antibody array was performed in total pool homogenate of mouse vWAT (*n* = 6 mice each group) fed with HFD or HFD + NV. Profiling Proteins: spotted cytokine antibodies; Reference Spot: spotted loading control; Control (−): PBS. Mean pixel spot density of each cytokine detected through antibody array showed in **g**. Data are expressed as means ± SD (*n* = 6 mice each group)

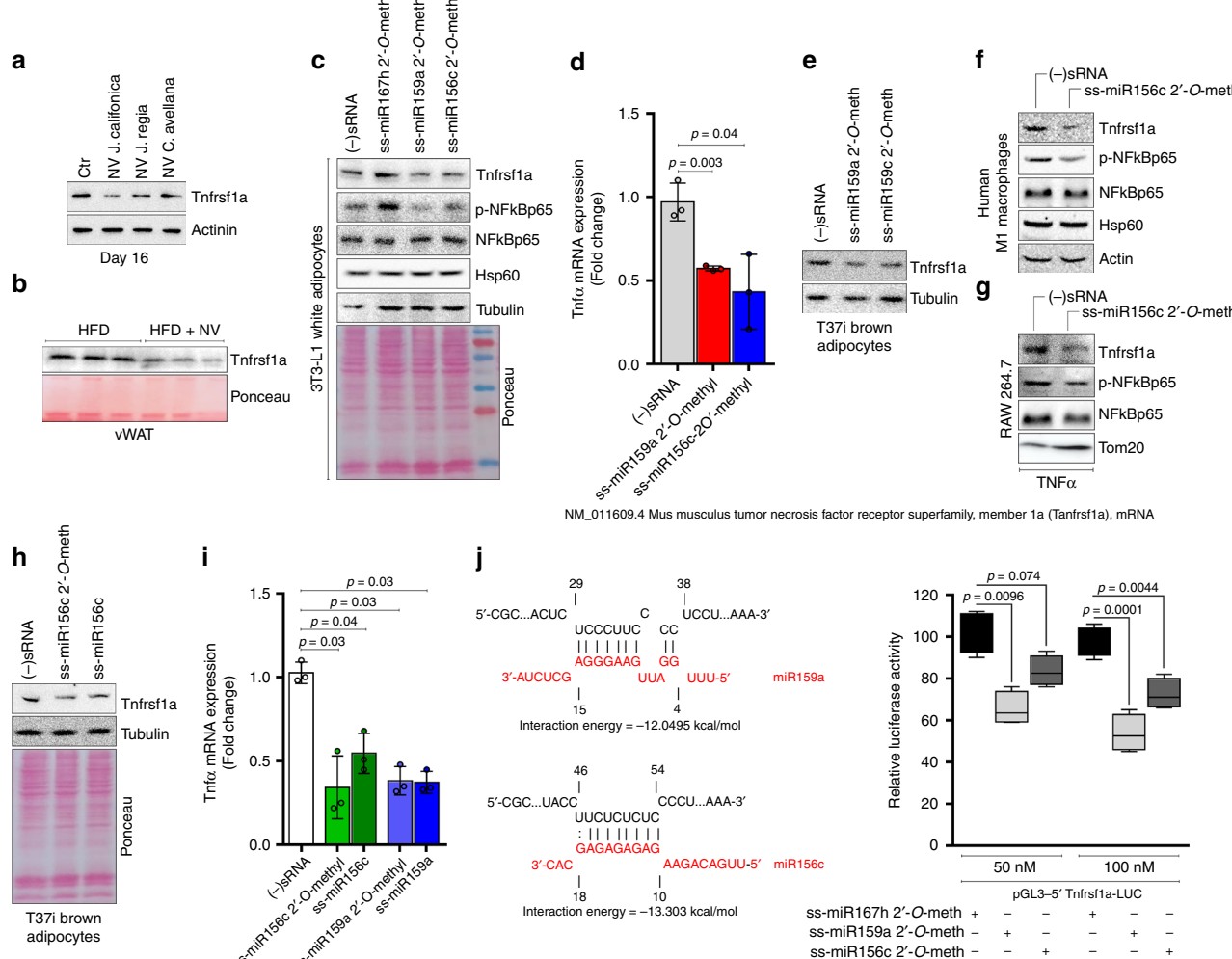

**Fig. 6 Synthetic ss-miR mimics for plant miR159a and miR156c target the Tnf signaling pathway. a, b** Tnfrsf1a protein levels in NV-treated adipocytes (**a**) and vWAT (**b**) of mice fed with high-fat diet (HFD) or HFD supplemented with NVs (HFD + NV). Uncropped images are shown in Supplementary Fig. 9. **c, d** Tnfrsf1a and p-NFkBp65 protein levels (**c**) and TNF-α mRNA expression (**d**) were analyzed in hypertrophic 3T3-L1 adipocytes transfected with single-stranded (ss) 2′-O-methylated miR159a or miR156c mimics. Transfection with ss-2′-O-methylated miR167h mimic or with a scramble small RNA [(−)sRNA] was used as negative control. Uncropped images are shown in Supplementary Fig. 9. **e–g** p-NFkBp65 and/or Tnfrsf1a protein levels were analyzed in differentiated T37i brown adipocytes (**e**), in primary human monocytes differentiated in M1 macrophages (**f**) and TNF-α-treated murine RAW 264.7 cells (**g**) transfected with ss-2′-O-methylated miR159a or miR156c mimics. Transfection with a scramble small RNA [(−)sRNA] was used as a negative control. Uncropped images are shown in Supplementary Fig. 9. **h, i** Tnfrsf1a protein levels (**h**) and TNF-α mRNA expression (**i**) were analyzed in T37i brown adipocytes transfected with single-stranded (ss) 2′-O-methylated, unmethylated miR156c, or miR159a mimics. Transfection with a scramble small RNA [(−)sRNA] was used as a negative control. Immunoblots reported are representative of three independent experiments giving similar results. Tubulin, actin, Hsp60, or Tom20 were used as loading controls. Uncropped images are shown in Supplementary Fig. 9. Data are expressed as means ± SD (n = 3). **j** HEK293 cells were transfected with the reporter construct (pGL3–5′Tnfrsf1a-LUC), containing the mouse 5′Tnfrsf1a region with predicted miR-binding sites (left panel) cloned upstream the luciferase gene, together with ss-2′-O-methylated miR159a, ss-2′-O-methylated miR156c mimics or negative control (ss-2′-O-methylated miR167h mimic). Firefly/Renilla luciferase activities were averaged (n = 10) and reported as residual activity of the respective transfections performed with negative control taken as 100%

human Tnf signaling pathway, was used. Synthetic ss-miR159a and ss-miR156c mimics were effective in decreasing Tnfrsf1a protein levels in adipocytes (Fig. 6c, e) and this resulted in reduction of p-NFkBp65 protein (Fig. 6c) and TNF-α expression (Fig. 6d). Human M1 macrophages and TNF-α-stimulated murine RAW 264.7 cells showed reduced Tnfrsf1a upon transfection with ss-miR156c mimic (Fig. 6f, g). As a consequence, reduction of p-NFkBp65 protein was also achieved in such cells (Fig. 6f, g). Notably, such results were recapitulated in T37i adipocytes transfected with ss-miR156c or ss-miR159a lacking 2′-O-methylation at 3′-end (Fig. 6h, i), suggesting that methylation does not affect biological activity of plant miR mimics.

Finally, to functionally validate the mouse sequence of Tnfrsf1a mRNA as a target region of 2′-O-methylated ss-miR159a and ss-miR156c mimics (Fig. 6c), the fragment with recognized sites was cloned upstream the luciferase gene in pGL3-vector, and luciferase activity was measured for 5′UTR-Tnfrsf1a-LUC in the presence of a negative control, 2′-O-methylated ss-miR159a, or ss-miR156c mimic. The results showed that the ectopic expression of synthetic ss-miR159a or ss-miR156c mimic inhibited the activity of the reporter construct bearing the 5′ UTR of Tnfrsf1a in HEK293 cells, when compared with negative control-transfected cells (Fig. 6j), demonstrating that Tnfrsf1a mRNA is the genuine target of ss-miR159a and miR156c mimics.

## Discussion

In the last years, the possible therapeutic use of plant miR mimics for modulating the animal gene expression machinery has raised increased attention. Plant miR mimics have been recently suggested to function as anti-inflammatory agents by targeting toll-like receptor 3 signaling in dendritic cells[18]. An anti-atherosclerotic function has been also observed through miR156a treatment that inhibits monocyte recruitment to inflamed endothelial cells[35]. Here we have focused our attention on the TNF-α signaling pathway as it takes center place in systemic inflammation and has a relevant role in the chronic low-grade inflammatory states characterizing aging as well as obese or diabetic individuals[36]. In the adipose tissue milieu of obese animals and humans, TNF-α is highly produced by hypertrophic adipocytes and infiltrated M1 macrophages[37]. TNF-α treatment reduces insulin sensitivity both in white and brown adipocytes, and obese mice lacking either TNF-α or its receptor show protection against developing insulin resistance[7]. For this reason, reducing adipose tissue inflammation via the targeting of TNF-α signaling may represent a promising therapeutic avenue. Herein, inspired by the widely reported anti-inflammatory properties of several plant foods, we firstly attempted at evaluating whether small RNA alone as well as delivered by NVs isolated from dried nuts were able to blunt adipocyte inflammation and insulin resistance in white and brown adipocytes. We found a reduction of a number of inflammatory cytokines in adipocytes and visceral adipose depots upon different pro-inflammatory conditions such as hypoxia, cellular hypertrophy, diet-induced obesity in association with the down-regulation of TNF-α inflammatory signaling pathway. In parallel, we found ameliorated metabolic parameters including improved glucose tolerance. TNF-α hyperproduction during infection and malignancy causes adipose tissue cachexia unraveling its important role in modulation of adipose tissue plasticity[38]. Decreased fatty acid uptake and lipogenesis in association with increased fat catabolism and inhibition of adipogenesis are the events ascribed to TNF-α activity in adipose tissue[38]. This could explain the increased adipose mass in mice fed with HFD and supplemented with nut NVs. Specifically, the adipocyte hypertrophy could be the result of the high flux of nutrients upon HFD that is not balanced by the cachectic action of TNF-α owing to the inhibitory activity of this inflammatory signal by NVs. This hypothesis is supported by a previous work showing that *Tnfrsf1a* knock-out mice have increased adipose mass compared to WT animals both upon normal feeding condition and HFD. Interestingly, similarly to HFD-fed mice supplemented with nut NVs, *Tnfrsf1a* knock-out mice also showed protection against HFD-induced inflammation in adipose tissue[39].

Our in vivo study represents a preventative approach that suggests the use of plant miRs to limit the development of insulin resistance and low-grade inflammation. Whether our approach can be used to treat established insulin resistance and inflammation remains to be investigated. However, in vitro experiments performed on hypertrophic and inflamed adipocytes revealed that plant miRs and nut NVs could reverse insulin resistance and inflammatory cytokines production, thus pointing to their possible therapeutic effects in vivo as well.

Through sRNA-sequencing and text data mining strategies, we identified conserved miR families in nuts and their NVs, and among these miR156c and miR159a were predicted to target Tnfrsf1a. miRs are double-stranded oligonucleotides but, although the use of duplex miR mimics is promising, they have potential drawbacks as therapeutic agents due to their limited bioavailability. Oligonucleotide antisense technology relies on single-stranded sequences of nucleotides that are complementary to RNA transcripts in the cells. In a general view, antisense

therapies were proposed to bind to an mRNA thus limiting the related protein availability. By exploiting the predicted anti-inflammatory potential of miR156c and miR159a, we developed an oligonucleotide antisense-like technology by synthetizing single-stranded miR mimics for miR156c and miR159a with 2′-O-methylation at 3′ end. As demonstrated by the luciferase assay, such miR mimics were able to target Tnfrsf1a gene transcript and lead to Tnfrsf1a protein down-regulation and inhibition of TNF-α signaling cascade both in inflamed adipocytes and macrophages.

TNF-α signaling has attracted attention since many years as several pathological conditions are characterized by deregulated engagement of Tnfrsf1a leading to acute and chronic inflammatory processes[36,40]. Targeting of TNF-α through recombinant antibody has found application in several inflammatory diseases such as neuroinflammatory and autoimmune diseases (e.g. rheumatoid arthritis, multiple sclerosis, and inflammatory bowel diseases) and cancer[12,41–43], but a proportion of patients may encounter adverse events[43,44]. As a consequence, such therapy is not appropriate for treating low-grade inflammatory states such as those observed in obesity or type 2 diabetes. In this context, treatment with single-stranded miR mimics for miR156c or miR159a may represent a promising therapeutic strategy to treat inflammatory diseases associated with enhancement of TNF-α signaling. Hence, by virtue of their efficient Tnfrsf1a targeting capacity, synthetic 2′-O-methylated miR156c and miR159a mimics may represent valuable therapeutics to reduce inflammation at least in the metabolic diseases associated with adipocyte dysfunction.

## Methods

**Cell culture, treatments, and transfections.** 3T3-L1 and RAW 264.7 cells were purchased from ATCC and grown in complete medium (Dulbecco's modified Eagle's medium (DMEM)) supplemented with 10% New Born Serum, 1% Pen/Strep mix, and 2 mM glutamine (Lonza Sales, Basel, Switzerland). T37i murine preadipocytes were kindly provided by Prof. Marc Lombes (INSERM U1185, Paris, France) and grown in complete medium (DMEM/F12) supplemented with 10% fetal bovine serum, 1% Pen/Strep mix, and 2 mM glutamine (Lonza Sales, Basel, Switzerland). 3T3-L1 and T37i cells were differentiated in adipocytes as previously described[22], with some modifications. In particular, 1 μM rosiglitazone was added to differentiation medium for 8 days. To induce a hypertrophic phenotype, differentiated adipocytes were maintained in complete medium (without hormones or drugs supplementation) for additional 8 days. Differentiated adipocytes (day 8) were treated with 1 mM cobalt chloride $CoCl_2$ (Sigma, St. Louis, MO, USA) or 200 ng/mL of recombinant mouse TNF-α (Sigma, St. Louis, MO, USA) for 16 h. All treatments (including NVs) were performed in serum-free media. Insulin signaling transduction capacity was analyzed by treating serum-starved adipocytes with 0.1 μM insulin in complete culture medium for 20 min.

Peripheral blood mononuclear cells were isolated from buffy coat preparations of anonymized healthy donors who gave their written informed consent to donate the non-clinically usable components of their blood for scientific research (according to the current Italian law). Monocytes were separated by using anti-CD14 monoclonal antibodies conjugated to magnetic microbeads (Miltenyi Biotec) according to the manufacturer's instructions. To get type I macrophages (M1) monocytes were separated and then suspended in complete medium and incubated for a further 5 days in 24-well plates at the concentration of $10^6$ cells/mL in the presence of 100 ng/mL granulocyte-macrophage colony-stimulating factor, as previously described[45].

Human HEK293 were purchased from American Type Culture Collection (ATCC) and cultured in DMEM supplemented with 10% (v/v) fetal bovine serum and 1% penicillin/streptomycin (Invitrogen) and transfected for luciferase assays with Lipofectamine 2000 (ThermoFisher). Ten μg/ml of apple or nut-extracted sRNA and 100 nM of single-stranded 2′-O-methylated miR159a, miR156c, miR167h, unmethylated miR156c, or unmethylated miR159a were transfected with Lipofectamine 3000 reagent (Thermofisher) for 48 h in serum-free media.

**NV purification and quantification.** NVs were isolated from whole homogenates of commercially available edible dried nuts (*Juglans regia, Juglans californica, Corylus avellana*) and fresh apple (*Malus domestica* "*Fuji*"). Three samples for each edible dried nut specimen were processed about 1 year after harvesting. Nut kernels and apple pulp were homogenized with Waring Blender (7012S Osaka) and then mixed with PBS (1:10 w/v). Successively, the homogenized samples were centrifuged at 20,000 × *g* for 15 min and the infranatant collected for centrifugation at

10,000 × *g* for 1 h. The supernatant was filtered by 0.45 μm pore filter and centrifuged at 16,500 × *g* for 1 h. The supernatant was then centrifuged at 110,000 × *g* for 70 min in a Type 70 Ti fixed angle rotor, the pellet was suspended in 1 ml PBS, filtered by 0.45 μm pore filter, and finally centrifuged at 16,500 × *g* for 10 min. The supernatant (NVs fraction) was collected for treatments, or miR extractions. To quantify NVs (μg/μL), an absorbance spectrum was initially performed on purified commercial exosomes (HansaBioMed Life Science, Lona, Switzerland) and a standard curve at absorbance peak (215 nm), which corresponds to absorbance peak of phospholipids, of known concentration of exosomes was defined (Supplementary Fig. 5).

**Triglyceride content**. Intracellular triglycerides content was evaluated in adipose cells by Oil Red O staining as previously described[46].

**Mice and treatments**. Mouse experimentation was conducted in accordance with accepted standard of humane animal care after the approval by relevant local (The University Animal Welfare Committee—OPBA, Tor Vergata University) and national (Ministry of Health, Legislative Decree No. 26/2014; European Directive 2010/63/UE) committees with authorization n°378/2017-PR.

C57BL/6J adult (2-month--old) female mice (purchased from ENVIGO, Italy) were randomly divided into three groups (*n* = 6 mice/group): (1) mice fed with normal diet (ND); mice fed with HFD (5.24 kcal/g among which 60% kcal from fat, 20% from protein, and 20% from carbohydrate). (2) Mice fed with HFD mixed with 5 μg/g food (about 20 μg NVs/day/mice) of *J. californica*-derived NVs (HFD + NV). The dietary treatments were maintained for 16 weeks and at the end of treatment blood from the facial vein was collected and adipose depots were explanted 6 h after fasting.

**Blood parameters and oral glucose tolerance test**. Glucose, cholesterol, alanine transaminase (Alt), and aspartate transaminase (Ast) were measured through the automatized KeyLab analyzer (BPCBioSed, Italy) using specific colorimetric assay kits (BPCBioSed) as previously described[47]. Oral glucose tolerance test was performed 8 weeks after dietary treatments by oral gavage (2 g of dextrose/kg body mass) in overnight fasted mice. Blood samples were collected from the tail vein at 0, 15, 30, 60, and 120 min after glucose loading, and the blood glucose level was immediately measured by a commercially available glucometer (Bayer).

**Sample preparation, total RNA, and small RNA extraction**. RNA extraction was performed on the edible part of nuts as described by Yockteng et al.[48], with some modifications. Briefly, dried nuts (50 g) were frozen in liquid nitrogen and then finely granulated with Waring Blender (7012 S Osaka). The frozen powder was transferred into a new tube and stored at −20 °C until use. Nut powder or pulp of fresh apple (2 g) was mixed with 10 mL of extraction buffer (100 mM Tris-HCl pH 9.5, 150 mM NaCl, 1% sodium lauroyl sarcosinate, 0.5% 2-mercaptoethanol) and strongly mixed for about 20 s in order to obtain a homogeneous solution. The solution was incubated overnight at room temperature in gentle agitation and then centrifuged (3,000 × *g* for 10 min) at room temperature. After centrifugation, the solution was stratified into three phases: a colorless upper phase containing fat particles, an interphase (400 μL) used for RNA extraction, and a lower phase containing impurities with higher weights. Total RNA purification was performed according to TRIzol reagent manufacturer's instruction (Sigma-Aldrich Co., MI, USA). Total RNA was resuspended in RNase-free water and treated with DNAse I (Promega, Italy) to remove contaminating DNA molecules. Small RNAs were extracted from total RNA by using a column-based isolation method (mirVana™ miR Isolation Kit; Ambion Inc.). RNA concentration and purity were evaluated by a Nanodrop ND1000 spectrophotometer (NanoDrop Technologies). The samples were stored at −80 °C up to processing.

**Constructing and sequencing of small RNA libraries, quality control analysis, and raw data filtering**. Individual indexed libraries were prepared from 1 μg of purified RNA using the TruSeq SmallRNA Sample Prep Kit (Illumina, USA) according to the manufacturer's instructions. Both glucose samples and final libraries were quantified by using the Qubit 2.0 Fluorometer (Invitrogen, Carlsbad, CA) and quality tested by Agilent 2100 Bioanalyzer RNA Nano assay (Agilent technologies, Santa Clara, CA). Libraries were then processed with Illumina cBot for cluster generation on the flow cell, following the manufacturer's instructions and sequenced on single-end mode 50 bp in multiplexing on HiSeq2500 (Illumina, San Diego, CA). The CASAVA 1.8.2 version of the Illumina pipeline was used to process raw data for both format conversion and de-multiplexing. Raw data sequence underwent quality controls analysis by using FastQC release 0.11.5 version [https://www.bioinformatics.babraham.ac.uk/projects/fastqc/] and MultiQC V 0.9 (ref. [49]). The Illumina small-RNA adapter was clipped using FASTX-Toolkit (http://hannonlab.cshl.edu/fastx_toolkit) and sequences shorter than 15 nt after trimming were removed and the remaining reads were trimmed on the 3p and 5p ends in order to remove low-quality bases. At the end of this procedure, BLASTn v2.2.30 with parameters "-task megablast, -perc_identity 100" was used to compare the remaining reads with the Rfam database (Rfam 11.0)[50]. This stringent step removed non-coding RNA (rRNA, tRNA, snRNA, snoRNA) and degraded fragments of mRNA.

**miR annotation from small RNA-seq**. Recently, several genome-wide miR annotation tools using small RNA-Seq have been developed to quantify the expression of annotated miRs and to predict novel ones. To perform this task on the selected edible dried nuts, we taken advantage of the miR-PREFeR pipeline[51], being a fast and versatile tool using expression patterns of miR, following the criteria for plant microRNA annotation[52] to accurately predict plant miRs from small RNA-Seq data samples. miR-PREFeR requires the genomes to be available locally and in the *fasta* format. Hence, we downloaded the *Juglans regia*[53] and *Corylus avellana* genomes from the NCBI Bioproject section. The *fastq* files containing the reads obtained through the sequencing experiments were converted in a *fasta* format characterized by a specific header, containing the number of reads for each sequence as required by the pipeline, using the provided *process-reads-fasta.py* script. A second script, *bowtie-align-reads.py*, was used to align the read files generated by *process-reads-fasta.py*. The scripts first generate the bowtie index files for the provided genome and then use eight threads to align the reads, generating as an output a *sam* alignment file. Unmapped alignments were filtered using SAMtools[54]. The *sam* alignment and *fasta* genome files were then used as input for the miR-PREFeR main program, using the parameters suggested by the authors for the miR prediction[51]. The output, consisting of several files containing reads count information for each mature and precursor predicted sequences, has been carefully analyzed for each of the sequenced nuts. The mature sequences were aligned using *blastn* against the Viridiplantae section of miRbase[55] to search for homologs of microRNA sequences and to verify the presence of novel, nuts-specific microRNAs. The analysis of *blastn* output, performed through in house written codes, revealed that nuts contain only miR that can be assigned to already known families and highlighted a different composition of the nuts miRNome in terms of miRs and relative abundances. Small RNA-seq from other dietary vegetable foods were download from Dietary miR database (http://sbbi-panda.unl.edu:5000/dmd/) and annotated in miR families to build Venn diagrams.

**Prediction of miR-mRNA interactions and evaluation of conserved node of TNF-α signaling pathway nodes in adipose cells**. IntaRNA v2.0 (ref. [56]) was used to identify the biological functions and regulatory relationships between dietary plant miRs and gene transcripts related to murine KEGG Tnf (*mmu04668*) signaling pathways and human TNFRSF1A gene transcript (ENSG00000067182). This algorithm outperforms other existing target prediction programs, considering the accessibility of target sites and the existence of a seed region for the interaction. In detail, the algorithm has been executed in the Exact mode (-m E), including both the seed constraint and target site accessibility in the calculations, and excluding low-energy interactions from the output (minimum free energy less than equal to −10 kcal/mol).

CyKEGGParser (http://apps.cytoscape.org/apps/cykeggparser) was used to evaluate the conservation rate of nodes mapping TNF-α signaling pathways in human adipose cells. The conservation rate was evaluated by setting gene expression threshold to 25 percentile of gene expression values download by BioGPS dataset (http://biogps.org).

**Plasmid constructions and luciferase reporter assay**. The Tnfrsf1a luciferase reporter construct was obtained by inserting the 5′UTR sequence of mouse Tnfrsf1a (NM_011609.4) (from +1 to +252) holding putative miR156c and miR159a-binding sites into the pGL3-control vector. The fragment was cloned by PCR amplification on mouse genomic DNA, using primer pairs with *Hind*III restriction enzyme sites in both forward and reverse primers (forward primer: 5′-ATCTTAAGCTTCGCTCTTGCAACACCACC-3′, reverse primer: 5′-CATCTA AGCTTCAGCAATTGACAACGCTCG-3′). PCR products were inserted into the pGL3-control vector in the *Hind*III site, upstream the Firefly luciferase coding region. The orientation of the cloned fragment was established by enzymatic digestion and confirmed by sequencing. This construct (5′UTR-Tnfrsf1a-LUC) was used at 200 ng in presence of Renilla luciferase reporter (10 ng), as an internal control[57]. For luciferase assays, transfection of HEK293 with single-stranded 2′-*O*-methylated miR159a, miR156c as well as with miR167h as negative control were performed using Lipofectamine 2000 Reagent (ThermoFisher) according to the manufacturer's instructions. Cells were plated in 24-well dishes at 60,000 cells per well 20 h before experiments, and each transfection was done in triplicate. Luciferase reporter assay was performed 24 h after transfection using a Dual Luciferase Reporter Assay System (Promega) according to the manufacturer's instructions on a 20/20n Luminometer instrument (Turner BioSystems).

**Transmission electron microscopy**. NVs pellets were dissolved in PBS containing 2 mM EDTA and 10 μL of each sample was deposited onto a 300-mesh copper grid for electron microscopy covered by thin amorphous carbon film (20 nm). After air drying, grid-mounted preparations were stained with 2% aqueous phosphotungstic acid solution, pH 7.3. Samples were observed under a FEI TECNAI 12 G2 Twin electron microscope (FEI Company, Hillsboro, OR, USA) at 120 kV equipped with an electron energy filter (Gatan image filter) and a slow-scan charge-coupled device camera (Gatan multiscan).

**Dynamic light scattering and electrophoretic light scattering analysis**. NVs pellets were subjected to measurement of size and ζ-potential thorough dynamic

light scattering and electrophoretic light scattering respectively by using a Nano-ZetaSizer (Malvern) apparatus equipped with a 5 mW HeNe laser. For size measurements the scattered light was collected at an angle of 173° and to get the size distribution of NVs the autocorrelation functions were analyzed by Mie theory by using the NNLS algorithm. To obtain NV number distribution the vesicle model implemented in the software of the instrument was employed and the refractive index was set to 1.38 according to what reported in the literature for mammalian extracellular vesicles[58]. For the measurement of the electrophoretic mobility, the phase analysis light scattering (PALS) implemented with the patented mode M3 (mixed mode measurement) was used in order to eliminate electro-osmotic effects and ameliorate accuracy and repeatability. The mobility ($\mu$) of the diffusing aggregates was then converted into a $\zeta$-potential through the Smoluchowski relation $\zeta = \mu\eta/\varepsilon$ ($\varepsilon$ = permittivity, $\eta$ = viscosity of the solution).

**Immunoblotting, cytokine antibody array, and ELISA assay.** Adipose cells and tissues were lysed in RIPA buffer (50 mM Tris-HCl, pH 8.0, 150 mM NaCl, 12 mM deoxycholic acid, 0.5% Nonidet P-40, and protease and phosphatase inhibitors) and proteins were loaded for SDS-PAGE followed by western blotting. Nitrocellulose membranes were incubated with primary antibodies (1:1000 dilution) and successively with the appropriate horseradish peroxidase-conjugated secondary antibodies. Immunoreactive bands were detected by a FluorChem FC3 system (ProteinSimple, San Jose, CA, USA) after incubation of the membranes with ECL Selected Western Blotting Detection Reagent (GE Healthcare, Pittsburgh, PA, USA).

Cytokine antibody array (R&D Systems Inc., Minneapolis, MN, USA) was used to detect the cytokine protein levels in 200 µg of a pool of total homogenate obtained from murine visceral adipose tissue. Cytokine content was determined through densitometric analysis according to the manufacturer's protocol.

Tnfα levels in sera and cell culture media were measured by murine and human TNF-α Quantikine ELISA kit according to the manufacturer's protocol (R&D Systems Inc.).

**RT-qPCR analysis.** For mRNA analysis, total RNA (3 µg) was retro-transcribed by using M-MLV (Promega, Madison, WI). qPCR was performed in triplicate by using validated qPCR primers (BLAST), Ex TAq qPCR Premix, and the Real-Time PCR (Applied Biosystem). For RT-qPCR analyses primers for TNF-α (forward: 5′-ATGGCCTCCCTCTCATCAGT-3′; reverse: 5′-CTTGGTGGTTTGCTACGAC G-3′), IL-6 (forward: 5′-GGATACCACTCCCAACAGACC-3′; reverse: 5′-GCCA TTGCACAACTCTTTTCTCA-3′), and IL-1β (forward: 5′-GCACTGGGTGGAA TGAGACT-3′; reverse: 5′-GGACATCTCCCACGTCAATCT-3′) were used. mRNA levels were normalized to actin mRNA, and the relative mRNA levels were determined through the $2^{-\Delta\Delta Ct}$ method.

miR retrotranscription and qPCR analysis were performed as reported by Gismondi et al.[59]. In brief, miR cDNAs were synthesized by using a reverse transcription kit specific for miR (miRCURY LNA Universal RT microRNA PCR, Synthesis Kit II; EXIQON), according to the manufacturer's guidelines. A synthetic spike-in control miR (UniSp6, EXIQON) was added to each retrotranscription reaction, in order to monitor quality and efficacy of the procedure, including the qPCR technique. The quantitation of each miR was performed by qPCR in a 10 µL reaction volume containing 10 ng cDNA, 50% SYBR green (Kapa SYBR Fast qPCR kit; Kapa Biosystems, Woburn, MA, USA), and 1 µL of the mixture of both miR-specific PCR primers (microRNA LNA PCR primer sets; EXIQON). qPCR temperature gradient was set as reported on EXIQON instruction manual, using a Biorad (IQ5) thermocycler. A melting curve was obtained for each amplification to verify the adequacy of the amplification. qPCR analysis was carried out on the miR reported in Supplementary Table 2, using 5S rRNA (whose specific primers were developed and designed by EXIQON Service on the basis of plant 5S rDNA sequences, including *Arabidopsis thaliana* [GenBank: AB073495.1]) and U6 small RNA[60,61] as an internal loading control, in plant and mouse extracts, respectively, to normalize of qPCR results (relative quantization). In fact, the amount of each miR was determined using the $2^{-\Delta\Delta Ct}$ formula and qPCR negative controls were always performed to validate the detection system.

**Flow cytometer analyses.** To monitor glucose uptake, adipose cells were serum-starved for 12 h and incubated with 100 µM of 2-NBDG (Invitrogen) and 0.1 µM insulin in complete culture medium. The cytofluorimetric analysis was performed by recording FL-1 fluorescence. To measure the intracellular expression of TNF-α, adipose cells were fixed, permeabilized with 70% ethanol, and incubated with PE-conjugated anti-mouse TNF-α (BD Biosciences, USA). Flow cytometry analyses were performed by Cytoflex (Beckman Coulter, USA) and analyzed by Cytexpert 1.2 software (Beckman Coulter, USA).

Cytoflex Beckman Coulter was also used for NVs analyses. Briefly, since the relative position of nanosized particles may differ depending on the cytometer's technological solutions and resolutions, we used a Megamix-Plus SSC standard microparticles kit (BIOCYTEX, France) for the NVs detection. Megamix-Plus kit contained nanosized FITC-A-conjugated standardized particles of different sizes: 100–160, 160–200, 200–240, and 240–500 nm. Gates for data acquisition of all vesicle samples are set accordingly to the manufacture's instruction. To evaluate the

RNA content, purified NVs were stained with 0.5 µM SYTO RNASelect, a membrane-permeant nucleic acid stain that selectively stains RNAs (ThermoFisher).

**Gas chromatographic-mass spectrometry.** Gas chromatographic-mass spectrometry analysis was performed by using a QP2010 GC-MS instrument (Shimadzu, Japan). Purified NVs (100 ng) were dissolved in 50 µL of ethyl acetate acidified with HCl 0.1% for 24 h in agitation, sonicated for 5 min twice, and vortexed for 2 h. Then, sample was centrifuged at 11,000g for 30 min. Supernatant was collected and dried out by a speed-vac system (Eppendorf AG 22331 Hamburg; Concentration Plus). The pellet was derivatized by resuspension with 15 µL of acidified ethyl acetate and 15 µL of the Methyl-8-Reagent (Thermo Scientific), according to the manufacturer's protocol. Two microliters of sample were injected in the chromatographic system set as follows: initial oven temperature at 60 °C for 5 min; then, at a rate of 4 °C/min, the oven reached 270 °C for 10 min. All system parameters and details were the same of those reported in Gismondi et al.[62], except that mass spectrum scanning range that varied from 40 to 800 $m/z$. Each molecule was identified by comparing its mass profile to those registered in the NIST (National Institute of Standard and Technology) Library 14 loaded on detection software of the instrument (Solution software). Only similarity values higher than 85% were considered acceptable. The identified molecules were analyzed by Human Metabolome Database (www.hmdb.ca) and the results were reported as Chemical Name and PubChem CID (Supplementary Table 1).

**Data and statistical analysis.** The accession number for the transcriptome profiling presented in this article are GSE32095 and GSE8044. Differential gene expression analysis was performed by GEO2R and Gene Ontology (GO) terms were analyzed by FunRich 3.0 tool and presented as biological processes. The results are presented as means ± SD. To compare the means of two groups unpaired Student's $t$-test was used. Nonparametric one-way ANOVA corrected with Kruskal–Wallis test was used for comparing the means of more than two groups. Differences were considered to be significant at $p < 0.05$ (GraphPad Prism 6). Through Venn diagrams (Venny 2.1) were defined the logical relations between small RNA-seq profiled in vegetable foods (http://sbbi-panda.unl.edu).

**Reporting summary.** Further information on research design is available in the Nature Research Reporting Summary linked to this article.

## Data availability

The datasets generated and analyzed during the current study are publicly available on BioProject (ID PRJNA553332). All other data that support the findings of this study are available in the Supplementary Information or from the corresponding author upon reasonable request. Full gel images are shown in the Supplementary Information.

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

## Acknowledgements

This work was supported by International Nut & Dried Fruit Council (INC, project No. 2016-R03), European Foundation for the Study of Diabetes (EFSD/Lilly 2017), and Italian Ministry of Health (GR-2018-12367588) to D.L.-B. F.I. was supported by an AIRC fellowship for Italy. A.M. and M.P. were supported by the STARBIOS2 European Union's Horizon 2020 research and innovation program under grant agreement No. 709517 oriented to promote Responsible Research and Innovation in biosciences.

## Author contributions

All authors contributed to the analysis and discussion of the results. K.A. and D.L.-B. supervised the research, wrote, and edited the manuscript; D.L.-B. conceptualized the

research and acquired funding; V.C., A.G., S.D.S., F.I., R.F., G.D.M., N.P., A. Marcone, G.M., A. Minutolo, F.T., S.S., S.C., M.P. and R.B. performed the experiments.

## Additional information

**Competing interests:** The authors declare no competing interests

