## [Peer Review File · Communications Biology]

Reviewers' comments:

Reviewer #1 (Remarks to the Author):

In this manuscript, Aquilano et al. investigate the anti-inflammatory actions of plant sRNAs. They demonstrate that Nut-derived nanovesicles and isolated Nut sRNAs reverse inflammation and improve insulin sensitivity in cultured adipocytes and NVs also improve glucose tolerance in a rodent model of obesity and insulin resistance. By sequencing several libraries of plant and nut sRNAs and they identified 12 conserved sRNA sequences. Amongst these were two miRNAs, namely miR156c and miR159a, which are predicted to target Tnfrsf1a. Finally, Aquilano et al. utilise Single-stranded mimics of miR156c and 159a to demonstrate that they can reduce TNFR1 protein expression. Overall, this is a very interesting, in particular the observation that *in vivo* anti-inflammatory action of nut-derived NVs is to improved glucose tolerance but this occurs alongside increased depot-specific adiposity. Although experimental approach was preventative, improved glycemic control does not appear to be as a result of reduced adiposity (weight gain) like some other anti-diabetic treatments. While this may not be clinically, appealing, the increase in Body weight and adipose depot sizes could be consistent with the notion that inflammatory signals are indeed critical for "limiting adipose tissue expansion" in obesity-linked insulin resistance.

Clearly the novelty in this report is not that targeting TNFR1 signalling improves both insulin sensitivity and HFD-induced glucose tolerance, But that :

- A) Plant derived miRNAs can target mammalian transcripts with anti-inflammatory action.
- B) Nut nanovesicles have anti-inflammatory activity that may be due to specific sRNAs.
- C) The approach used here, is an excellent example of interdisciplinary science; bringing together plant biology, transcriptomics & bioinformatics, 'nanotechnology' nanovesicles to potentially treat human disease.

Specific comments

1. In figure 1c, 4b and 6c- The immunoblots of NFkBp65 are puzzling. My understanding is that during inflammation, it is the intracellular localisation of p65 that is regulated and it is also post-translationally modified during inflammation. Hence, the suggestion that TNFalpha, CoCl2 treatment or adipocyte hypertrophy increases total p65 protein expression (suggested here as positive control for activation of inflammation) is not consistent with this understanding. Please clarify.
2. In figure 1i – the data representing glucose uptake is not common and appears to be a graphical representation but with no statistical information. With limited information the effect of sRNAs seems extremely modest and does not seem reliable. Consider presenting this data as for Fig4e.
3. Using a non-candidate screening approach the authors have identified 12 conserved plant miRs. Two of these were identified as miR156c and miR159a, but what was the identity of the remaining 10 conserved plant miRs what do they target? How do these relate to those identified in isolated nut nanovesicles (NVs)?
4. Are NV-induced changes in adipose depots a reflection of increased adipocyte hypertrophy or hyperplasia? What was the impact on lipid accumulation in the livers?
5. Do HFD-fed TNFa-KO or HFD fed TNFR1-KO mice have increased adipose mass too relative to HFD-fed wt mice?
6. The *in vivo* study is a preventative approach, which may be appropriate for preventative strategies

with dietary supplements. However, whether this approach can be used to reverse established insulin resistance remains unclear.

Minor points

1. Consider reorganising the order in which the results are presented. Begin with NV activity, followed by nut & plant sRNAs screening (and identification of two miRs followed by confirmation of putative targets and siR mimics).
2. The introduction ends with the conclusion "The data here presented point to the use of plant miR-based single-stranded oligonucleotides for the treatment of chronic low-grade inflammatory states such as those observed in ageing and obesity." However no ageing-related models or data are presented.

Reviewer #2 (Remarks to the Author):

In this manuscript, Aquilano et al. report beneficial the role of the single-stranded miR-156c and 159a in inflammation and adipocyte insulin sensitivity. The authors found the presence of conserved plant miR159a and miR156c in dried nuts. By bioinformatics calculation, Tnfrsf1a was predicted as the potential target. Treating with NVs including miR-159a and miR-156c reduced Tnfrsf1a protein in adipocytes. In addition, synthetic single-stranded miRNAs showed same effect in cultured cells and animals. Although the anti-inflammatory role of ss-miRs is interesting, there are several issues that the author may wish to address.

Comments:

1. The authors showed that the small RNAs isolated from dry nuts limit inflammatory response and enhance insulin sensitivity in hypertrophic adipocytes. However, only in vitro data were shown, so is it possible to quantify the exact RNA concentration that might work in vivo? So how many nuts should human/mouse eat each day to achieve the same effect.
2. As the authors mention in figure 2, they chose the most conserved miR-156c and miR-159a for further investigation, it is well known that these two miRs are also highly expressed in varies plants that human daily consume, does those plants, like green veggies also exert their beneficial role through these two miRs?
3. Insulin tolerant test should also performed to further strengthen the improved insulin sensitivity.
4. Is there any possible mechanism of the edible NVs in animals? As we known that the gastric acid might break the membrane structure of NVs?

Reviewer #3 (Remarks to the Author):

There is a lot of nice hard work in this manuscript. However, a prevailing concern is the types of "negative" controls that are used throughout the manuscript. In the first batch of experiments that authors look at how small RNA isolated from nuts reduce cytokine expression. Each nut RNA appears effective- a better 'negative" control would be RNA derived from another type of plant. The scrambled small RNA is not sufficient. If this is a nut specific effect another batch of RNA from a different plant should not exhibit this phenotype.

The section on small RNA sequencing is rigorous and sufficient. However, history has shown that computational target predictions do not tell a story that can be easily translated into biological effects.

The plant NVs section looks adequate but again- as a control another plant NVs could be isolated and used for comparisons (see below).

As in section 1 NV from another plant should be used to monitor the TNG levels and glucose uptake in adipocytes.

In the inflammation assays with obese mice another NV supplemented diet should be included to make sure it is specific to the nut derived NVs. Again use another plant outside of nuts.

The section of synthetic ss-miR mimics needs to look at the role of the 2'O methylated form of the mimics. If you remove the 2'O group does this render the RNA ineffective? Why is this a plant specific effect?

Again, tons of work here but the controls and not sufficient. If the authors look back at a paper by Mlotshwa et al. 2015 they will see that the plant RNA by itself without the 'active" miRNA had a modest effect. The authors here have not done adequate controls for the effects of plant miRNAs that are off target and have not adequately address the specificity of the nut derived NVs.

RESPONSE TO REVIEWERS

Reviewer #1:

We thank this Reviewer for having considered our manuscript very interesting and saying that “*the approach used here, is an excellent example of interdisciplinary science; bringing together plant biology, transcriptomics & bioinformatics, ‘nanotechnology’ nanovesicles to potentially treat human disease*”. Following are our responses to his/her criticisms.

Response to Specific comments

1. ***In figure 1c, 4b and 6c- The immunoblots of NFkBp65 are puzzling... Please clarify.*** We agree with this reviewer that NFkB activity is regulated by its intracellular redistribution into the nuclei and post-translational modifications (i.e. phosphorylation). The NFkBp65 antibody (Santa Cruz Biothechnology) we used specifically recognizes the phospho-active form. Moreover, we had checked the basal form of NFkB that, as it remained unchanged, we decided to not report in the figure. We apologize for not having previously well specified this issue throughout the manuscript and having generated such confusion. We have now specified that the NFkB p65 corresponds to the phospho-active form (p-NFkBp65) both in the figures and in the main body of the manuscript.

2. ***In figure 1i – the data representing glucose uptake is not common and appears to be a graphical representation but with no statistical information.*** OK. We have changed the graphical presentation of Fig. 1i.

3. ***Using a non-candidate screening approach the authors have identified 12 conserved plant miRs. Two of these were identified as miR156c and miR159a, but what was the identity of the remaining 10 conserved plant miRs what do they target? How do these relate to those identified in isolated nut nanovesicles (NVs)?*** OK. We have provided the identity of the remaining 10 miR families (miR-166, -167, -171, -172, -319, -396, -398, -399, -403, -482) in Table 1 and none of these are predicted to target nodes of TNF signaling according to computational analysis carried out with IntaRNA v2.0. Finding other signaling pathways putatively targeted by such conserved miRs within the entire mammalian transcriptome is challenging and behind the scope of the present work that was instead centered on TNF pathway. Along with miR156c and miR159a, 4 out of 12 of the conserved miRs were also detected in nut NVs (i.e. miR166i, miR167-5p, miR396c, miR482b) (see suppl. Fig 3).

4. ***Are NV-induced changes in adipose depots a reflection of increased adipocyte hypertrophy or hyperplasia? What was the impact on lipid accumulation in the livers?*** Through H&E staining of subcutaneous adipose tissue (new Supplementary Fig. 4a) and Oil Red-O staining of NVs-treated adipocyte (new Supplementary Fig. 4b), we demonstrated the occurrence of adipocyte hypertrophy, as an increased dimension of lipid droplets. Accordingly, increased level of intracellular triglycerides was detected in T37i adipocytes treated with nut NVs. We did not evaluate histochemically the lipid content in the livers as they appeared morphologically similar in terms of weight and color between HFD and HFD+NV.

5. ***Do HFD-fed TNFa-KO or HFD fed TNFR1-KO mice have increased adipose mass too relative to HFD-fed wt mice?***

This aspect has been properly discussed and referenced. In brief, we have mentioned a published study (PMID:19477937, i.e. ref. 39) in which it was demonstrated that TNFR1-KO mice show increased adipose mass compared to WT animals both upon normal feeding condition and high fat diet. We have also highlighted that, in line with our results, the increased adiposity of such mice is associated with an anti-inflammatory effect of TNFR1 knockdown.

6. ***The in vivo study is a preventative approach, which may be appropriate for preventative strategies with dietary supplements. However, whether this approach can be used to reverse established insulin resistance remains unclear.***

We agree with the referee's comments and stated in the discussion that our experimental approach and the obtained results highlight the preventive anti-inflammatory action of plant miRs. However, we are confident that plant miRs could be effective in reversing established states of insulin resistance and inflammatory cytokine production. This assumption is supported by results in which plant small RNAs were able to mitigate insulin resistance and inhibit the expression of inflammatory cytokines in the *in vitro* model of hypertrophic adipocytes. This evidence has been appropriately discussed and whether plant miRs could be effective in treating established insulin resistance and chronic inflammatory states is part of our future research (see ll. 276-281).

Response to minor points

1. Consider reorganising the order in which the results are presented. Begin with NV activity, followed by nut & plant sRNAs screening (and identification of two miRs followed by confirmation of putative targets and siR mimics). We have attempted at reorganizing the results description according to the Reviewer's suggestion but we weren't able to do this as our scientific strategy as well as the rationality of our work resulted significantly affected.

2. The introduction ends with the conclusion "The data here presented point to the use of plant miR-based single-stranded oligonucleotides for the treatment of chronic low-grade inflammatory states such as those observed in ageing and obesity." However no ageing-related models or data are presented. OK. We have modified this sentence.

Reviewer #2:

We have appreciated the Reviewer's comments and addressed his/her concerns as follows:

1. ...is it possible to quantify the exact RNA concentration that might work in vivo? So how many nuts should human/mouse eat each day to achieve the same effect. Our study suggests that plant small RNAs could be used at a pharmacological dose in order to prevent obesity-related insulin resistance and low-grade inflammatory states. Actually, for the *in vitro* experiments we used 10 ug/ml of small RNAs, which correspond to 0,4 g of nuts. Considering that mice have about 2 ml of blood, to reach the same concentration, the diet of mice should be supplemented with 0,8 g of nuts/day. It means that for human (about 6 L blood) a dose of 2.4 Kg of nuts has to be consumed to achieve the same effects.

2. ...does those plants, like green veggies also exert their beneficial role through these two miRs? Also according to the suggestions of Reviewer 3, we have now added new experiments in which we demonstrated that small RNAs from another vegetable food, i.e. fresh apple (*M. domestica*), increase glucose uptake and exert anti-inflammatory effects at a minor extent than small RNA from nuts (new Fig. 1f and Fig. 1i). Moreover, we show that apple NVs are ineffective in modulating inflammatory cytokine production (see new Fig. 4a). As apple NVs display minor content of miRs than overall the analyzed nut species (see Suppl. Fig. 3c), the results obtained suggest that the anti-inflammatory potential can be dependent on plant miRs abundance.

3. Insulin tolerant test should also performed to further strengthen the improved insulin sensitivity. We had not measured insulin levels but we have only performed the OGTT. Unfortunately, due to bio-ethic issues we cannot repeat the HFD+NV experiments on animals.

4. Is there any possible mechanism of the edible NVs in animals? As we known that the gastric acid might break the membrane structure of NVs? Plant NVs have been reported to be highly resistant to gastric acid as referenced in the results section (new ref. 32).

Reviewer #3:

We thank this Reviewer for his/her critical review and for the suggestions that in our opinion have improved the quality of our manuscript. In summary, we have now added new experiments in which we demonstrated that small RNAs from another vegetable food (i.e. fresh apple, *M. domestica*) increase glucose uptake and exert anti-inflammatory effects at a minor extent than small RNA from nuts (new Fig. 1f and Fig. 1i). Moreover, we show that apple NVs are ineffective in modulating inflammatory cytokine production (see new Fig. 4a). As apple NVs display minor content of miRs than overall the analysed nut species (see Suppl. Fig. 3c), the results obtained suggest that the anti-inflammatory potential can be dependent at least in part on plant miRs abundance. Finally, we have included new experiments with unmethylated miRs. Following is the detailed response to the points raised by the Reviewer.

... a better 'negative' control would be RNA derived from another type of plant... We have now added new experiments in which we show that apple small RNAs, even though at lesser extent than small RNAs from nuts, are able to inhibit cytokine production and increase glucose uptake in adipocytes (see new Fig. 1f and 1i); see also Reviewer 2, point 2).

The plant NVs section looks adequate but again- as a control another plant NVs could be isolated and used for comparisons ... OK. We have repeated the experiments using apple-derived NVs and found that they are not effective in inhibiting cytokine production (see new Fig. 4a) in line with the lower content of miRs, including miR156c and miR159a, with respect to nut NVs (Suppl. Fig. 3c).

In the inflammation assays with obese mice another NV supplemented diet should be included to make sure it is specific to the nut derived NVs. Again use another plant outside of nuts.

Our new results show that apple NVs are ineffective in inhibiting pro-inflammatory cytokine production *in vitro*. Therefore, also due to important ethic issues, we could not carry out new experiments on animals by using NVs from another plant.

The section of synthetic ss-miR mimics needs to look at the role of the 2'O methylated form of the mimics...

OK. We have repeated experiments with unmethylated miR156c and miR159a and found similar results (Tnfrsf1a and cytokine production downregulation), suggesting that methylation does not affect biological activity of the transfected plant miRs (new Fig. 6h and Fig. 6i).

Why is this a plant specific effect? It is not a plant specific effect; actually, unmethylated miRs are able to exert the same effects as methylated miRs (new Fig. 6h and 6i). The 2' O methylation, that is naturally present in plant small RNA, has been recommended for therapeutic use of small RNA, as it may confer higher stability *in vivo* (new ref. 33).

Again, tons of work here but the controls and not sufficient. If the authors look back at a paper by Mlotshwa et al. 2015 they will see that the plant RNA by itself without the 'active' miRNA had a modest effect. The authors here have not done adequate controls for the effects of plant miRNAs that are off target and have not adequately address the specificity of the nut derived NVs.

In their paper, Mlotshwa et al., inspired by the higher stability of methylated plant miRs with respect to unmethylated mammalian miRs, designed mammalian methylated anti-tumor miRs to increase their stability.

They demonstrated that such modified mammalian miRs, like plant miRs, are taken up by the digestive system and maintain their anti-tumour activity. They used a total plant RNA spiked with such mammalian/methylated miRs and total plant RNA alone as control to demonstrate the specificity of the mammalian/methylated anti-tumour miRs. Plant RNAs alone were not able to exert an anti-tumour activity, thus demonstrating that methylation per se is not antitumoral but the anti-tumor effect depends on specific miR nucleotide sequence. Our plant small RNA pool as well as NVs naturally contain “active” miR156c and miR159a, hence, we could not reproduce the control used by Mlotshwa et al. The specificity of miR156c and miR159a, in our opinion, has been demonstrated by transfecting methylated miR167 and small RNA from apple that do not show significant effect on the TNFalpha signaling. Moreover, the specificity of miR sequence in targeting Tnfrsf1a has been also demonstrated by using 2'-O-methylated miR and unmethylated miR.

Reviewers' comments:

Reviewer #1 (Remarks to the Author):

RESPONSE TO REVIEWERS

Reviewer #1:

We thank this Reviewer for having considered our manuscript very interesting and saying that "the approach used here, is an excellent example of interdisciplinary science; bringing together plant biology, transcriptomics & bioinformatics, 'nanotechnology' nanovesicles to potentially treat human disease". Following are our responses to his/her criticisms.

Response to Specific comments

1. In figure 1c, 4b and 6c- The immunoblots of NFkBp65 are puzzling... Please clarify. We agree with this reviewer that NFkB activity is regulated by its intracellular redistribution into the nuclei and post-translational modifications (i.e. phosphorylation). The NFkBp65 antibody (Santa Cruz Biothechnology) we used specifically recognizes the phospho-active form. Moreover, we had checked the basal form of NFkB that, as it remained unchanged, we decided to not report in the figure. We apologize for not having previously well specified this issue throughout the manuscript and having generated such confusion. We have now specified that the NFkB p65 corresponds to the phospho-active form (p-NFkBp65) both in the figures and in the main body of the manuscript.

Reviewer: Thank you for this clarification. It is disappointing that the p65 protein levels were not included as the specific protein loading control in these blots (incorrectly referred to the authors as "basal form of NFkB"). Likewise, all phosphosite-specific blots should be routinely accompanied by loading blot for that specific protein. It is clear that this approach has not been followed in relation to all the phospho-site-specific blots (both for p-NFkBp65 and pAKT). IF total NFkBp65 protein and total AKT protein blots have been performed, it is not clear why they are not included or provided for the review. IF they cannot be included then this limitation should be noted in the discussion of the manuscript.

2. In figure 1i – the data representing glucose uptake is not common and appears to be a graphical representation but with no statistical information. OK. We have changed the graphical presentation of Fig. 1i.

Reviewer: Neither the figure legend, not methods sections have been edited to reflect the methodology used for glucose uptake measurements. What was the duration of the various incubations with 2-NBDG? With Insulin? When, how much and for how long were 16 day old adipocytes treated with each sRNA? Was this prior to the 2-NBDG and insulin treatments? What was the number of replicates (both for biological and experimental)?

3. Using a non-candidate screening approach the authors have identified 12 conserved plant miRs. Two of these were identified as miR156c and miR159a, but what was the identity of the remaining 10 conserved plant miRs what do they target? How do these relate to those identified in isolated nut nanovesicles (NVs)? OK. We have provided the identity of the remaining 10 miR families (miR-166, -167, -171, -172, -319, -396, -398, -399, -403, -482) in Table 1 and none of these are predicted to target nodes of TNF signaling according to computational analysis carried out with IntaRNA v2.0.

Finding other signaling pathways putatively targeted by such conserved miRs within the entire mammalian transcriptome is challenging and behind the scope of the present work that was instead centered on TNF pathway. Along with miR156c and miR159a, 4 out of 12 of the conserved miRs were also detected in nut NVs (i.e. miR166i, miR167-5p, miR396c, miR482b) (see suppl. Fig 3).

Reviewer: I am satisfied with the amended table 1 and the additional information provided here.

4. Are NV-induced changes in adipose depots a reflection of increased adipocyte hypertrophy or hyperplasia? What was the impact on lipid accumulation in the livers? Through H&E staining of subcutaneous adipose tissue (new Supplementary Fig. 4a) and Oil Red-O staining of NVs-treated adipocyte (new Supplementary Fig. 4b), we demonstrated the occurrence of adipocyte hypertrophy, as an increased dimension of lipid droplets. Accordingly, increased level of intracellular triglycerides was detected in T37i adipocytes treated with nut NVs. We did not evaluate histochemically the lipid content in the livers as they appeared morphologically similar in terms of weight and color between HFD and HFD+NV.

Reviewer: Neither of these new data specifically answer the question posed. Specifically no quantification is provided for H&E histology and no information is provided for how invitro TAG accumulation is normalized. Establishing whether increased lipid accumulation is driven by increased adipocyte cell number or larger adipocytes is an important issue to address here.

5. Do HFD-fed TNFa-KO or HFD fed TNFR1-KO mice have increased adipose mass too relative to HFD-fed wt mice?

This aspect has been properly discussed and referenced. In brief, we have mentioned a published study (PMID:19477937, i.e. ref. 39) in which it was demonstrated that TNFR1-KO mice show increased adipose mass compared to WT animals both upon normal feeding condition and high fat diet. We have also highlighted that, in line with our results, the increased adiposity of such mice is associated with an anti-inflammatory effect of TNFR1 knockdown.

Reviewer: I beg to disagree. Firstly, ref 39 relates to mice lacking both TNFR1 and TNFR2 (RKO). They also suggest that "Obese RKO mice were markedly insulin resistant, suggesting that intact TNFR signaling is not required for the effect of obesity to impair glucose metabolism" These authors also clearly state that "Of note is the observation that genetic ablation of both R1 and R2 results in phenotypes that are distinct from mice lacking only one receptor or TNF itself." There are also additional reports that have been omitted. Hence it is critical that the this area is properly/accurately discussed and referenced.

6. The in vivo study is a preventative approach, which may be appropriate for preventative strategies with dietary supplements. However, whether this approach can be used to reverse established insulin resistance remains unclear.

We agree with the referee's comments and stated in the discussion that our experimental approach and the obtained results highlight the preventive anti-inflammatory action of plant miRs. However, we are confident that plant miRs could be effective in reversing established states of insulin resistance and inflammatory cytokine production. This assumption is supported by results in which plant small RNAs were able to mitigate insulin resistance and inhibit the expression of inflammatory cytokines in the in vitro model of hypertrophic adipocytes. This evidence has been appropriately discussed and whether plant miRs could be effective in treating established insulin resistance and chronic

inflammatory states is part of our future research (see ll. 276-281).

Reviewer: I am satisfied with the amendments made to the discussion.

Response to minor points

1. Consider reorganising the order in which the results are presented. Begin with NV activity, followed by nut & plant sRNAs screening (and identification of two miRs followed by confirmation of putative targets and siR mimics). We have attempted at reorganizing the results description according to the Reviewer's suggestion but we weren't able to do this as our scientific strategy as well as the rationality of our work resulted significantly affected.

Reviewer: Okay.

2. The introduction ends with the conclusion "The data here presented point to the use of plant miR-based single-stranded oligonucleotides for the treatment of chronic low-grade inflammatory states such as those observed in ageing and obesity." However no ageing-related models or data are presented. OK. We have modified this sentence.

Reviewer: Okay.

Reviewer #3 (Remarks to the Author):

The authors have done the additional experiments- I respect the amount of effort put into this manuscript

REVIEWER #1 RESPONSE

We thank this Reviewer for the valuable suggestions regarding the methodology and the discussion. Following is our response to all the concerns raised.

1. It is disappointing that the p65 protein levels were not included as the specific protein loading control in these blots

OK. We have now added p65NFkB and Akt protein levels as specific loading controls. We really apologize for not having provided these immunoblots in the previous version of the work.

2. Neither the figure legend, nor methods sections have been edited to reflect the methodology used for glucose uptake measurements. Ok. We have edited the methods and figure legends as suggested.

What was the duration of the various incubations with 2-NBDG? With Insulin? Ok. See p. 17, ll. 542-545).

When, how much and for how long were 16 day old adipocytes treated with each sRNA? Was this prior to the 2-NBDG and insulin treatments?

Ok. See p. 11, ll. 326-327 and p. 17, ll. 542-545.

What was the number of replicates (both for biological and experimental)?

Ok. See p. 18, ll. 583-584.

4... no quantification is provided for H&E histology and no information is provided for how in vitro TAG accumulation is normalized. Establishing whether increased lipid accumulation is driven by increased adipocyte cell number or larger adipocytes is an important issue to address here.

OK. We have now calculated the diameter of lipid droplets of adipocytes using ImageJ (see new Fig 4a, right panel). Moreover, we have now normalized Oil red O absorbance to the protein concentration (see new Fig. 4b). Methods have been revised accordingly (see p. 12, ll. 360-364).

5. Ref. 39 relates to mice lacking both TNFR1 and TNFR2 (RKO)... Hence it is critical that the this area is properly/accurately discussed and referenced.

Ok. We have improved the discussion and appropriately referenced it according to the reviewer's suggestions (see pp. 9-10, ll. 271-282).

REVIEWERS' COMMENTS:

Reviewer #1 (Remarks to the Author):

N/A